# Community-Based Interventions to Reduce Child Stunting in Rural Guatemala: A Quality Improvement Model

**DOI:** 10.3390/ijerph18020773

**Published:** 2021-01-18

**Authors:** Michel Juarez, Carlos Dionicio, Neftali Sacuj, Waleska Lopez, Ann C. Miller, Peter Rohloff

**Affiliations:** 1Center for Research in Indigenous Health, Wuqu’ Kawoq|Maya Health Alliance 2a Avenida 3-48 Zona 3, Barrio Patacabaj, Tecpán, Chimaltenango 04006, Guatemala; michel@wuqukawoq.org (M.J.); carlos@wuqukawoq.org (C.D.); neftali@wuqukawoq.org (N.S.); waleska@wuqukawoq.org (W.L.); 2Department of Global Health and Social Medicine, Harvard Medical School, 641 Huntington Ave, Boston, MA 02115, USA; ann_miller@hms.harvard.edu; 3Division of Global Health Equity, Brigham and Women’s Hospital, 75 Francis Street, Boston, MA 02115, USA

**Keywords:** stunting, malnutrition, health disparities, indigenous populations, rural populations, quality improvement, community health worker

## Abstract

Rural Guatemala has one of the highest rates of chronic child malnutrition (stunting) in the world, with little progress despite considerable efforts to scale up evidence-based nutrition interventions. Recent literature suggests that one factor limiting impact is inadequate supervisory support for frontline workers. Here we describe a community-based quality improvement intervention in a region with a high rate of stunting. The intervention provided audit and feedback support to frontline nutrition workers through electronic worklists, performance dashboards, and one-on-one feedback sessions. We visualized performance indicators and child nutrition outcomes during the improvement intervention using run charts and control charts. In this small community-based sample (125 households at program initiation), over the two-year improvement period, there were marked improvements in the delivery of program components, such as growth monitoring services and micronutrient supplements. The prevalence of child stunting fell from 42.4 to 30.6%, meeting criteria for special cause variation. The mean length/height-for-age Z-score rose from −1.77 to −1.47, also meeting criteria for special cause variation. In conclusion, the addition of structured performance visualization and audit and feedback components to an existing community-based nutrition program improved child health indicators significantly through improving the fidelity of an existing evidence-based nutrition package.

## 1. Introduction

Chronic child undernutrition continues to be a global problem in low- and middle-income countries. The most commonly used indicator for chronic undernutrition is stunting, or low length/height-for-age. From 2000 to 2019 global stunting prevalence rate declined more than 10%. However, gains were unequally distributed, with some countries—such as India and China—exhibited strong positive trends, while in other countries nearly 1 in 2 children remain affected by stunting [1].

Guatemala has the highest stunting rate in Latin America and 6th highest worldwide, and the annual rate of reduction has averaged only 0.45% over the last 20 years, despite intensive investments and efforts [2,3,4]. Repeatedly, ambitious national political milestones to reduce stunting dramatically—such as the 24% reduction proposed by the National Secretary of Food and Nutritional Security for 2006–2016—have not been met [5]. The intractability of stunting can be explained in part by the complex ways that child nutrition intersects with other determinants of well-being. Stunting disproportionately occurs in rural agricultural communities in Guatemala, where most inhabitants are indigenous Maya. In these communities, rates of stunting routinely exceed national levels by 50% or more [6]. Food insecurity and poverty are also more common, with almost 60% of the population living in poverty and 16% living with severe food insecurity [6,7]. Finally, equitable access to primary healthcare is limited, with the Government of Guatemala abruptly ending its initiative to expand rural access in 2014, a public financing concession in theory guaranteed under terms of the Peace Accords of 1996 which concluded the country’s long civil war [8,9]. 

In addition—and particularly relevant to the rural and indigenous communities where need is greatest—the need for adaptive and flexible implementation are also an important part of the picture. This may help to explain why, for example, even recent evidence-based packages of stunting interventions aligned with the international Scaling Up Nutrition movement priorities have failed to achieve the same impact in Guatemala as in other peer countries [10]. For example, in a recent scoping review of nutrition literature from Guatemala, lack of attention to contextual and implementation details were identified as important barriers [4,11]. Similarly, a qualitative review of nutrition scale-up activities highlighted how deficiencies in frontline capacity impeded implementation of central nutrition policy decisions [12]. Along these lines, global guidelines highlight improving supervisory and support structures for front-line health workers as a core best practice which is often poorly implemented [13]. Finally, centralized design decisions and lack of meaningful mechanisms for engaging local actors in defining and adapting programs undermines sustained community engagement and violates the principles of the right to prior consultation embedded in the International Labor Organization’s Indigenous and Tribal People’s Convention, of which Guatemala is a signatory [14].

Rapid-cycle quality improvement (RCQI) methods are emerging as an important technique for designing and monitoring health systems interventions, which use pragmatic non-experimental techniques to introduce and rapidly evaluate interventions to improve system performance and quality [15,16]. Use of RCQI is rapidly expanding in high-income countries, but its use in low- and middle-income settings remains limited, especially in community-based settings. However, the technique is ideally suited to the types of complex and ongoing contextual adaptations that characterize community-based interventions in low-resource settings. They also may help bridge the “implementation gap” facilitating review of process weak points and meaningful audit and feedback for frontline health workers [17]. Given the literature reviewed above, we hypothesized that a RCQI approach focused primarily on improving the supervision of frontline nutrition workers could be an effective strategy to improve the impact of otherwise evidence-based rural community nutrition programs. Here we describe the use of this methodology to improve outcomes from a rural nutrition intervention in Guatemala and describe the impact of the strategy on stunting prevalence.

## 2. Materials and Methods

### 2.1. Participants and Setting

The Maya Health Alliance is a nonprofit primary care organization working in five departments of central Guatemala, where it provides healthcare services, primarily in rural agricultural indigenous Maya communities. A major programmatic focus is community nutrition programs to improve child growth outcomes in communities affected by high rates of child stunting. Particular elements of these programs are evidence-based and align with Guatemala’s national strategy for chronic malnutrition [3]. We have previously described the clinical elements of these programs in detailed as well as detailed their modest impact on children’s diet quality and growth outcomes (Appendix A) [11,18].

In this manuscript, we describe the long-term implementation of this package in one rural Maya community, Xik’injuyu’ (a pseudonym), which is located in the southwestern piedmont region of Guatemala. This is small agricultural community of roughly 350 households, most of indigenous K’iche’ Maya descent. Most families work in a mixture of subsistence agriculture and as day laborers on large sugar cane, coffee, rubber, and banana plantations. Maya Health Alliance began implementation of their nutrition program (Table 1) in this community in October 2014, and all households with children under 5 years of age were eligible to participate, with rolling recruitment and exit from the program based on this age criterion. At the time this program began, roughly 70% of families in the community lived in poverty (less than $2 USD/day), and 60% reported household food insecurity.

This study was deemed to be a quality improvement intervention and therefore exempt from human subjects research review by the Maya Health Alliance Institutional Review Board (Protocol Number WK 2017 008, 15 October 2017). The Revised Standards for Quality Improvement Reporting Excellence were used to draft this manuscript [19].

### 2.2. Description of Improvement Intervention

In October 2017, after noting slow but modest improvements in community stunting rates, we formed a performance improvement team at Maya Health Alliance. The team included a supervising nutritionist, staff pediatricians, representative nutrition staff (trained community health workers and auxiliary nurses), and an informatics specialist. Together, the team noted inconsistent feedback mechanisms for frontline workers, and underutilization of monitoring and evaluation functions in Maya Health Alliance’s electronic health record system (www.openmrs.org) as key opportunities for improvement. The team proposed several specific activities, including more structured feedback to frontline nutrition workers, automated task lists, and regular visualization of performance data produced by mining electronic health record data (Figure 1, Table 1). Importantly, in keeping with institutional philosophy as well as the core principles of quality improvement, the team placed emphasis on collaborative strategies designed to explore and amplify “what works” while also learning from and deprecating less successful approaches. Within this framework, the “supervisor” has the skills to perform higher-level data processing/visualizing and serves as a sounding board to help frontline workers analyze their own successes and challenges, but does not primarily enforce a performance standard.

After the start of the initiative, the team met every month to review data and to modify to interventions using a “Plan-Do-Study-Act” methodology [15]. An informatics specialist extracted process and clinical outcomes data from the electronic health record, which was presented in graphical form to all members of the team by the team nutritionist, who served as the overall lead for the project. A timeline of selected interventions and modifications over the time course of the improvement initiative is given in Figure 2. Monitoring of outcomes and the first regular improvement team meetings began in January 2018. 

### 2.3. Data Collection and Measures

Data for this improvement study was from two sources. First a baseline needs assessment survey conducted in October 2014 at the start of the Maya Health Alliance community collaboration was used to generate aggregate descriptive sociodemographic statistics. This survey included data on household structure, occupation, education, and maternal and child health were collected. A 24-h dietary recall to assess child diet quality was performed, using the technique outlined by the World Health Organization [20]. Household food insecurity was assessed using the Household Food Insecurity Access Scale [21]. Household probability of living below the national poverty line was estimated using the Quick Poverty Score, a validated tool referenced to the National Standards of Living Survey [22]. Anthropometric measurements (weight, height/length) were collected by trained technicians using standard techniques and Z-scores were calculated using the WHO Child Growth Reference Standards [23,24]. 

Second, during the improvement intervention, routine clinical data was extracted from the electronic health record by an informatics specialist using SQL search routines. This data was used to construct a series of core indicators which were monitored in monthly improvement team meetings (Table 2). Most of this data was summarized quarterly; when more than one observation for an individual was available within a given quarter, the observation closest to the midpoint of the quarter was used. All available data in the electronic health record for each quarter was analyzed. Data was extracted from the beginning of the nutrition needs assessment and intervention (October 2017) through December 2019. Visualization of both baseline performance and evolution of indicators after the start of the improvement initiative began in January 2018. 

### 2.4. Analysis

We performed statistical analyses in Minitab 18 (State College, PA, USA), Stata 14 (College Station, TX, USA), and R version 3.6.3 (Vienna, Austria). Baseline sociodemographic and clinical characteristics of participating children and families were summarized using mean and standard deviation (for parametric continuous variables); median and interquartile range (for nonparametric continuous variables) using Shapiro-wilk test to assess normal distribution of data; or percentages (for categorical variables).

To assess the impact of the improvement initiative, we used statistical quality control charts, including a proportion chart (for stunting prevalence) and a mean/standard deviation chart (for mean height/length-for-age Z score) [25]. On quality control charts, data are graphed over time, and upper and lower control limits (±3 SD, UCL and LCL) are plotted around the measure of interest. Special cause rules, which are similar to the concept of ‘statistical significance’ in traditional methods, are then applied to determine if the trend in the data are unlikely to be due to chance by comparing a trend to a baseline performance period [15]. For this analysis, the baseline performance period was October 2014–December 2017, and we applied the following commonly used special cause rules: Rule 1, 1 point > 3 standard deviations from central line; Rule 2, 9 or more points either above or below the central line; Rule 3, 6 or more consecutive points either all going up or down; Rule 4, 14 consecutive points alternating up and down across central line [26].

Time points for each indicator included all available data for that indicator, meaning that individual children’s data were often present at multiple time points. This data structure can lead to autocorrelation between time points, as can other important time-variable factors such as seasonality. To control for autocorrelation, we therefore performed sensitivity analysis using the generalized estimating equations function in Stata (xtgee command) for the proportion of stunted children (binomial family, logistic link function) or mean height/length-for-age Z-score (Gaussian family, identity link function) pre- and post-intervention. For both regression analyses, we used an exchangeable correlation structure to account for intra-subject correlations. The generalized estimating equations approach was most appropriate for our purposes, given our primary interest in the overall group trend, rather than an analysis of individual child-level responses and covariates.

## 3. Results

### 3.1. Baseline Characteristics of Participants and Caregivers

At the time of program initiation, a door-to-door household survey (October 2014) collected baseline sociodemographic and clinical data on 165 children (125 households) under 5 years of age. Selected characteristics are summarized in Table 3. Overall, families in the program community were poor, with a median poverty score corresponding to a nearly 80% probability of living below the national poverty line. 50% of households were also food insecure, and although a majority of heads-of-household work as agricultural day-laborers, only 29% had access to land and grew their own food for home consumption. In terms of child health, more than half of all surveyed children were stunted at baseline, and dietary indicators were poor, with around half meeting World Health Organization standards for diet diversity and meal frequency.

### 3.2. Monitoring of Process Improvement Indicators

The primary hypothesis of our improvement team was that improved supervision and feedback to frontline nutrition workers, assisted by regular visualization of performance data through audits electronic health record data, would improve the delivery of nutrition program elements and improve child outcomes. To monitor this, we used run charts to follow process indicators for worker-client contact frequency (proportion of children receiving recommended growth monitoring (Figure 3B) and delivery of nutrition components (proportion of children receiving micronutrient supplements (Figure 3C). Both indicators showed improvements during the intervention period.

Furthermore, to ensure that any observed changes were not simply a function of decreasing program size (e.g., failing to maintain community-wide coverage by recruiting new children), we monitored the number of new children enrolled in the program quarterly. As can be seen in Figure 3A, the number of new enrollees remained stable through the project period after initial program start-up in October 2014. 

Program enrollment data was tracked throughout, but data elements required to monitor process indicators were only put in place at the beginning of the improvement intervention. For this reason, data on enrollment is shown over the life of the program (Figure 3A), whereas data on process indicators (Figure 3B,C) is shown from the start of the improvement intervention.

### 3.3. Improvements in Child Growth Outcomes

In addition to process indicators, we also assessed the proportion of children who were stunted (height/length-for-age Z-score < −2) from the start of the nutrition program in October 2014 until January 2020, using a proportion control chart. We used the time period from October 2014 through December 2017 to calculate the baseline performance period, taking improvement team meetings in January 2018 as the start of the improvement intervention. The mean pre-intervention proportion of stunting was 42.4%, decreasing to 30.6% in the intervention period (Figure 4). Special cause according to Rule 1 was observed in quarter 4 2018 and again in quarters 2 and 3 2019. Special cause according to Rule 2 was obtained in quarter 4 2019.

We also examined mean height/length-for-age Z-scores for all children enrolled in the program using the same time limits with a mean and standard deviation (X-bar and S) control chart [15]. The mean pre-intervention height/length-for-age Z-score was −1.77, decreasing to −1.47 in the intervention period (Figure 5A). Special cause according to Rule 1 was observed in quarter 4 2018 and again in quarters 2 2019. Special cause according to Rule 2 was obtained in quarter 4 2019. The variance around the mean Z-score also decreased significantly in the intervention, as evidenced by special cause on the standard deviation control chart in quarter 2 2019 (Rule 3) (Figure 5B). This change in variance was driven largely by a reduction in the number of children with extreme low values of height/length for age z-score, as can be seen by comparing dot plots of individual Z-scores from a pre-intervention time point with high variance (quarter 4 2015) with a post-intervention time point with lower variance (quarter 3 2019) in Figure 6.

### 3.4. Sensitivity Analysis 

Given the repeated-measures panel structure of the data, with individual children included in multiple time points, we used generalized estimating equations to estimate the impact of the improvement intervention while accounting for intra-subject correlation. In these analyses the proportion of stunted children declined 17% (95% CI 3 to 31%, *p* = 0.02) in the improvement period when compared to the baseline. Similarly the mean height/length-for-age Z-score improved by 0.06 (95% CI 0.003–0.12, *p* = 0.04).

We also considered the possibility that the first quarter of data may have been an outlier (qualitatively worse stunting and Z-scores than the immediately subsequent quarters, Figure 4 and Figure 5). To address this, we repeated control chart and generalized estimating equations analyses excluding this quarter. Run charts were qualitatively unchanged, and special cause was still obtained (results not shown). The improvement in stunting obtained by generalized estimating equations was also similar (16%, 95% CI 2 to 30%, *p* = 0.02).

## 4. Discussion

The nutritional status of young children in agricultural communities remains a vexing national problem, and the intractability of stunting over many years in rural Guatemala to evidence-based technical nutrition interventions demands explanation. One important factor is that most nutrition interventions in Guatemala have remained fairly narrowly focused on proximal (“nutrition specific”) determinants of child health without adequate attention to broader policies that address broader (“nutrition sensitive”) determinants, such as poverty alleviation and food sovereignty [27,28]. An additional dimension is that attention to program adaptability and supervisory support in nutrition programs are likely major determinants of impact. Qualitative work has demonstrated how lack of clear “marching orders” for frontline actors have limited the impact of prior efforts to scale nutrition policy; furthermore, despite a large technical nutrition literature in Guatemala, evidence of best practices in program implementation are largely missing [4].

Against this background, here we present a case study of a comprehensive nutrition program in one rural Guatemala community, taking advantage of more than 5 years of administrative and programmatic data to evaluate changes in implementation strategy. In particular, using a quality improvement analytical framework, we investigate how the addition of new support and feedback tools for frontline workers (Figure 1, Table 1) impacted child nutrition indicators over and above standard evidence-based technical nutrition interventions (Figure 2). The impact of these audit and feedback strategies was marked, with significant improvements in process indicators, reflecting both improvements in delivery of nutrition components and contact frequency with frontline workers (Figure 3A,B). These process improvements were accompanied by a decrease in the proportion of children with stunting from 42.4 to 30.6% (Figure 4), findings which were robust to our sensitivity analyses. For comparison, the rate of decline in stunting from 1965 to 2014 nationally has been 17% [29].

Our case study provides detailed implementation evidence for the effectiveness of augmenting supervisory (audit and feedback) support for frontline nutrition workers in rural Guatemala. This evidence is of national importance, as implementing institutions and policymakers work to improve child growth indicators and to understand what factors predict program success or failure [4]. In addition, our study articulates with a larger global literature on audit and feedback, most of which however is situated in high income settings [17,30]. In low- and middle-income countries settings a few studies have documented the impact of audit and feedback for improving health outcomes, but most have targeted physicians or centralized care facilities [31,32,33,34,35,36,37]. A few studies, however, have evaluated the use of audit and feedback for frontline workers. For example, a performance review and mentoring intervention in Ethiopia improved both work volume and the ability of extension workers to diagnose common illnesses [38]. In Mozambique and Uganda, an intervention that included the use of smart phones to provide direct voice connections to supervisors as well as decision support and performance feedback also improved the management of common childhood illnesses [39]. Our study adds to this small body of literature, providing a specific focus on Latin America and on child nutrition. Importantly, recently released global guidelines on best practices for frontline health worker programs emphasize the need to improve supervisory structures [13]. 

In addition, it is important to emphasize that the implementation of audit and feedback described here occurred within an institutional context committed to collaborative, non-punitive relationships between supervisors and frontline workers that emphasized problem solving and deemphasized performance standards. Although this implementation culture is a prerequisite for effective quality improvement work, it is by no means the norm in supervisory culture in Guatemala. Recently, for example, another team in Guatemala has described difficulties in quality improvement work occasioned by authoritarian leadership and limited teamwork [40]. Similarly, the abrupt defunding of rural primary healthcare services by the Government of Guatemala in 2014 was in part occasioned by inflexible, bureaucratic insistence of unrealistic performance measures which made frontline workers and organizations responsible for structural factors well outside their scope of control [8,9]. 

Our cases study has several important limitations. First, it represents programmatic data from one rural community in Guatemala; since local demographic and ecological factors vary widely throughout the country, our findings may not be applicable to other community partnerships in other parts of the country. Furthermore, as our interest was primarily the group-level improvement, we did not perform an individual child- or household-level analysis and therefore our analysis cannot speak to the impact of individual-level covariates. Next, as a quality improvement study designed to assess overall program performance, our analysis cannot provide insight into the relative importance or efficacy of any of the individual program modifications made during the intervention period. Finally, as a non-experimental study using routine clinical data extracted from an electronic health record and without the benefit of a control group, we cannot completely exclude the possibility that systematic biases in beneficiary enrollment or data collection, or confounding by secular trend or intra-subject autocorrelation, although our findings were robust to controlling for these factors in a time-series generalized estimating equation.

## 5. Conclusions

In conclusion, we evaluated the addition of performance visualization tools and one-on-one audit and feedback for frontline health workers leading a community-based nutrition program in a region of rural Guatemala with a high prevalence of stunting. These tools improved the delivery of key program elements, and they led to a sustained reduction in the prevalence of child stunting. Our cases study provides important new evidence into “what works” in nutrition programming in rural Guatemala and should encourage other implementers and policy makers to consider similar supervisory support mechanisms when designing programs. Future work that we plan to undertake includes evaluating the differential effectiveness of this audit and feedback approach in distinct regions of Guatemala and across different institutional contexts.

## Figures and Tables

**Figure 1 ijerph-18-00773-f001:**
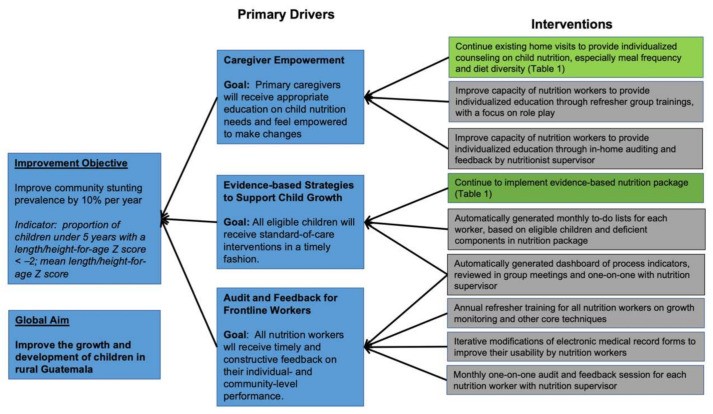
Driver Diagram for Nutrition Intervention Improvement Initiative. Areas of focus for improvement are given in gray boxes in the right-hand column. Ongoing inputs from the existing nutrition intervention are indicated in green boxes.

**Figure 2 ijerph-18-00773-f002:**
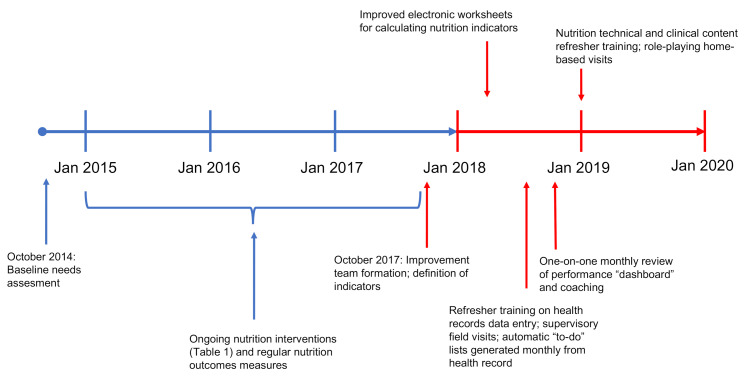
Timeline of Nutrition Program and Improvement Interventions. The pre-intervention period (nutrition program implementation) is given in blue, and the improvement initiative (augmentation of the nutrition program with improved supervision, feedback, and digital tools) is given in red.

**Figure 3 ijerph-18-00773-f003:**
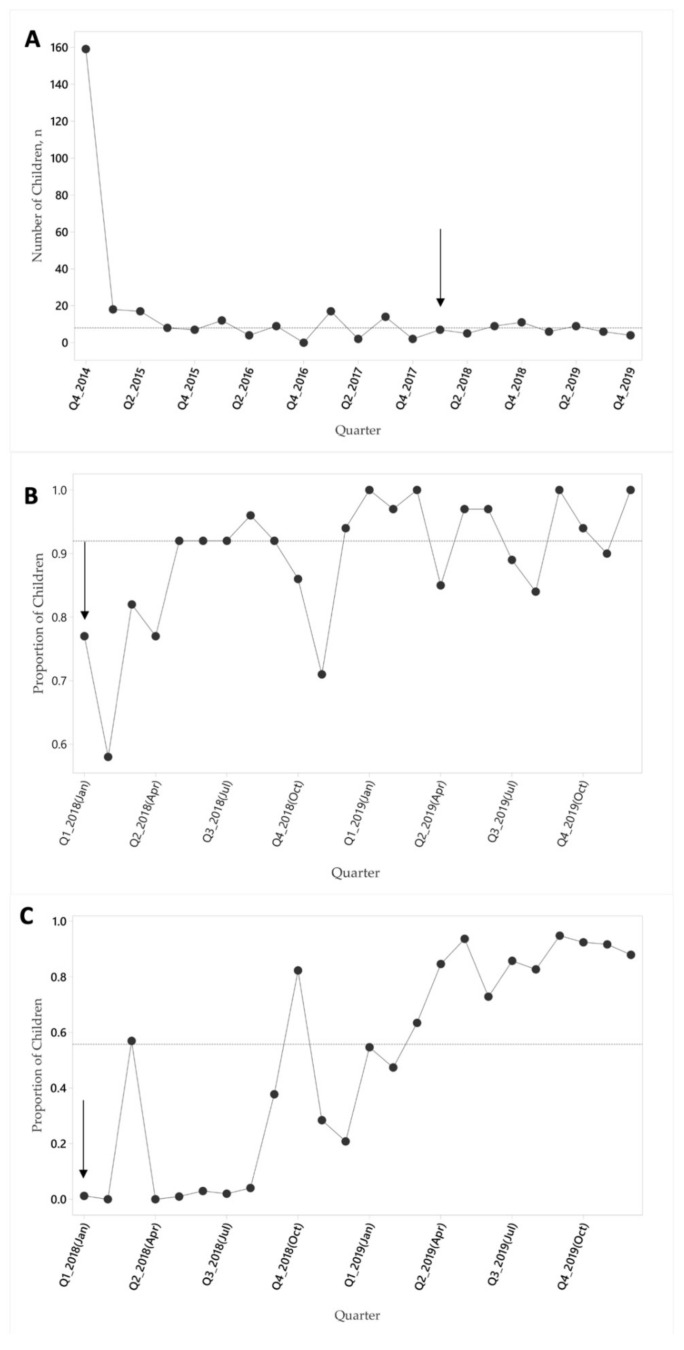
Run charts of total children recruited quarterly in community program (**A**), proportion receiving recommended monthly growth monitoring (**B**), proportion receiving recommended monthly micronutrient supplements (**C**). Arrow indicates the start of the improvement intervention and dashed line indicates the median for each indicator.

**Figure 4 ijerph-18-00773-f004:**
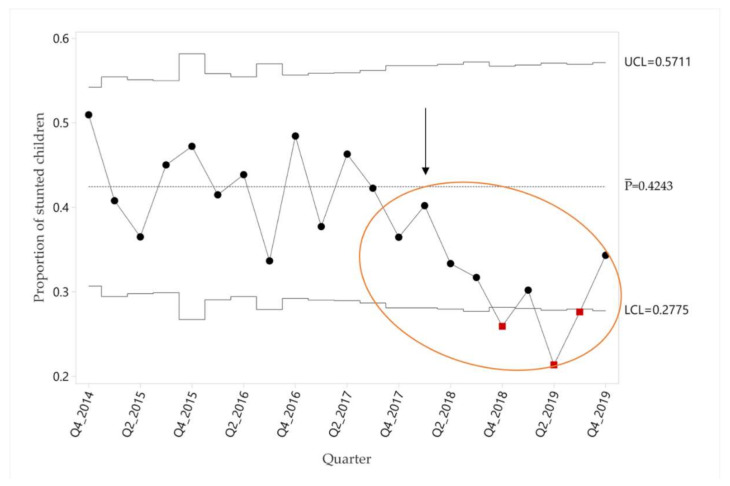
Control chart of the quarterly proportion of stunted children in community program. Upper (UCL) and lower control limits (LCL), and baseline mean proportion ($\bar{P}$) are shown. Arrow indicates the start of the improvement intervention. Red points indicate obtaining Special Cause Rule 1 and circle indicates Special Cause Rule 2.

**Figure 5 ijerph-18-00773-f005:**
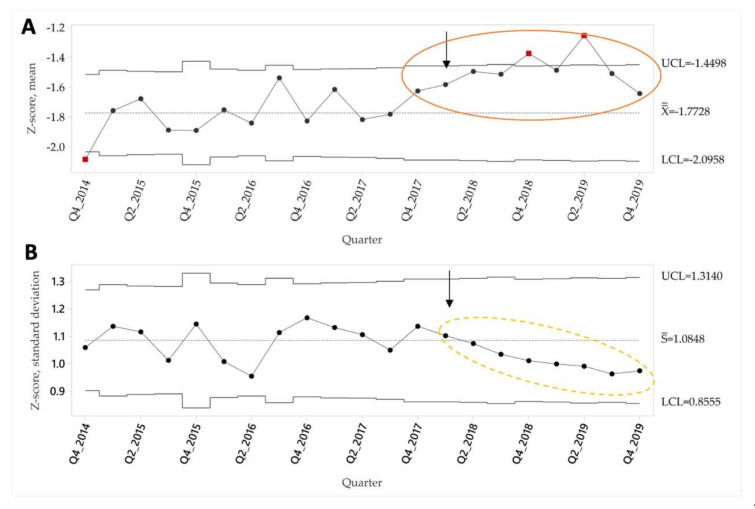
Mean (**A**) and Standard Deviation (**B**) Control Chart: Height/Length for age Z-score. Upper (UCL) and lower control limits (LCL), baseline mean height/length for age z-score ($\bar{X}$), and baseline mean standard deviation ($\bar{S}$) are shown. Arrow indicates the start of the improvement intervention. Red points indicate obtaining Special Cause Rule 1, solid circle indicates Special Cause Rule 2, and dashed circle indicates Special Cause Rule 3.

**Figure 6 ijerph-18-00773-f006:**
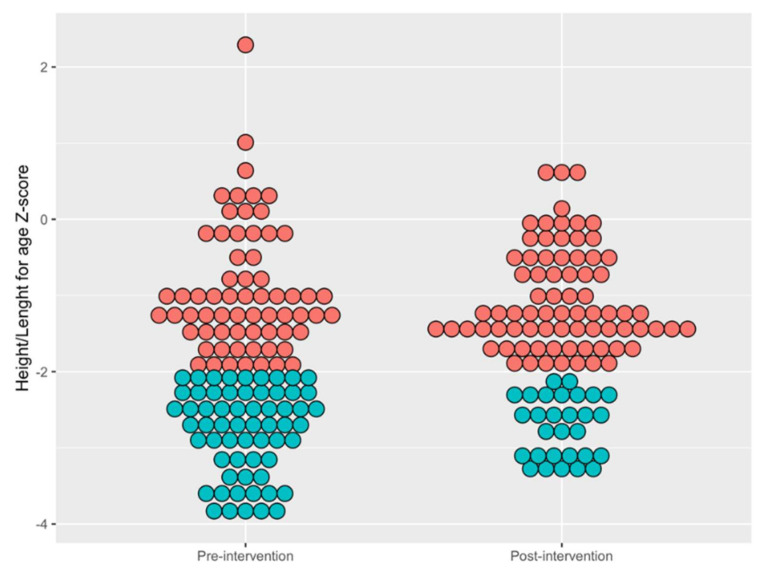
Dot Plot of Height/Length for age Z-score of Children measured pre and post-intervention. Red dots indicate individual children who were not stunted (height/length for age z-score > −2), and blue dots indicate children who were stunted (height/length for age z-score ≤ −2). Data shown for one pre-intervention quarter (quarter 4 2015) and one post-intervention quarter (quarter 3 2019).

**Table 1 ijerph-18-00773-t001:** Description of selected improvement strategies.

Component	Rationale	Example Interaction
Refresher group trainings, focused on role-play of worker-caregiver interactions	Small groups of frontline workers simulate challenging clinical encounters, allowing for peer feedback on body language, counselling strategy, and other skills	A frontline worker demonstrates a skilled approach to caregivers’ concerns that they “don’t have time” to peers
Individual audit/feedback of home visits by nutrition supervisor	Supervision of home visits, if conducted respectfully, can provide opportunities to reflect and improve interpersonal skills	Supervisor notes frontline worker struggling with a loud home environment, together they work to strategize how best to delivery content when multiple children are present
Automatic To-Do lists for pending growth monitoring, nutrition supplements, and clinical visits generated from electronic health record	Manually tracking program tasks is time-consuming and prone to error, leading to beneficiaries and program components being missed	Frontline worker has a busy work day and doesn’t notice that a child with severe malnutrition misses their appointment. After checking the task list, they notice the pending item and make a special home visit.
Automatic dashboard of process indicators (proportions of children receiving recommended growth monitoring, clinical visits, and nutrition supplements), with monthly individual review with supervisor	Visualization of process indicators, with the help of a supportive supervisor, can help identify “what’s working” and “what’s not working” and leading to creative problem solving	Supervisor points out that frontline worker’s home visit numbers were much higher than in previous months. Frontline work notes that phone calls the night before planned visits were effective at ensuring caregivers were prepared, and resolves to adopt this strategy going forward.

**Table 2 ijerph-18-00773-t002:** Core Improvement Indicators.

**Outcome Indicators**	**Description**
Prevalence of stunting	Proportion of children under 5 years of age with a height/length-for-age Z-score of less than—2
Height/length for age Z-score	Quarterly mean of height/length-for-age Z-score measures on all children under 5 years of age
**Process Indicators**	**Description**
Micronutrient Delivery	Proportion of eligible children with monthly documentation of micronutrient dosing in electronic health record
Growth Monitoring	Proportion of eligible children with monthly visits where weight and height were taken and documented in electronic health record
Program Enrollment/Stability	Quarterly number of children under 5 years enrolled in the community nutrition program

**Table 3 ijerph-18-00773-t003:** Selected baseline demographic and clinical characteristics of participants (October, 2014).

Characteristic ^1^	Value
**Household characteristics**	
Household size, *n*	4 (3, 5)
Children under 5 years in home, *n*	1 (1, 2)
Grows food for home consumption, %	29
Raw poverty score ^2^	32 (25, 37)
Moderate or severe food insecurity, %	50
**Child characteristics**	
Child age, months	23 (16, 33)
Female sex, %	54
Height/length-for-age Z score	−2.15 (−2.91, −1.39)
Weight-for-age Z score	−1.25 (−1.95, 0.46)
Stunted, % ^3^	52
Meets minimum dietary diversity, % ^4,5^ (n = 71)	58
Meets minimum meal frequency, % ^4,6^	56
Diarrhea in last two weeks, %	36
Fever in last two weeks, %	29
Respiratory symptoms in last two weeks, %	58
**Caregiver characteristics**	
Head of household works as agricultural day-laborer, %	71
Paternal education, primary school or less	57
Maternal education, primary school or less	74
Maternal pregnancies, n	2 (2, 4)

^1^ Values are given as medians (interquartile range) or raw percentages, as appropriate. ^2^ A raw score of 30–39 corresponds to an 77% probability of living below the national poverty line; lower scores are indicative of more poverty. ^3^ Height/length-for-age Z-score below −2. ^4^ These World Health Organization dietary quality indicators are only calculated for children under 2 years of age (n = 86). ^5^ At least 4 distinct food groups consumed in 24 h. ^6^ At least 2 solid meals (or milk feeds for non-breastfed infants) in 24 h for breastfed infants 6–8 months old, 3 for breastfed infants 9–23 months, and 4 for non-breastfed infants 6–23 months.

## Data Availability

The data presented in this study are openly available in Dataverse at https://doi.org/10.7910/DVN/XYEARX.

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
