# Peer review of "Community-Based Interventions to Reduce Child Stunting in Rural Guatemala: A Quality Improvement Model"

_ijerph, 2021, doi:10.3390/ijerph18020773_

Round 1

Reviewer 1 Report

This manuscript provides a detailed study of the impact of an improvement intervention on an existing rural health program in Guatemala. As the authors note, there have been numerous health programs addressing chronic childhood malnutrition in Guatemala for decades, with various but often limited results. The authors hypothesize that part of this lack of sustained impact may be due to a narrow focus on proximal determinants while ignoring the importance of support and empowerment of frontline workers. To address this, the authors implemented an intervention program focused on providing more supervision and feedback for frontline workers.

The value of the study comes from the length of time in the community of study, with baseline data collected in 2015 and monitoring data collected every quarter after. As the authors note, other publications have demonstrated the impact of the Maya Health Alliance’s on various health measures in other communities which help to understand the effectiveness of the program and reasons for the improvement intervention.

While I cannot comment on the data analysis, the authors present a clear discussion of the data collected, the analysis process, and the evidence for significant changes in process measures (proportion receiving growth monitoring and micronutrient supplements) as well as outcome measures (proportion of children with stunting).

My main concerns regarding the paper have to do with the description of the intervention program and the mechanisms through which the intervention fosters reductions in children’s malnutrition.

The intervention component is described briefly in different parts of the manuscript, but the actual content and importance of each component is not always clear. Most often the intervention is described as feedback and audit tools, which includes “automatic “worklists” of deficient tasks; performance dashboards; one-on-one feedback sessions with a supervisor” (lines 288-289). These components are also described as both supervisory support as well as a mechanism for empowerment, yet the mechanism through which these components impact the process or outcome measures is not clear.

For example, does the visualization of performance data and the worklist help frontline workers improve the number of households visited each month, or to track the administration of components to each child in the program? Did the number of household visits or number of households visited increase during the training program? What did the monthly one-on-one supervisory meetings consist of? Are these for support, to talk about difficulties frontline workers experience, or management of data and reinforcement of expectations?

Along with this, the use of metrics and performance data as a mechanism for empowerment may need to be more developed. Metrics and clear performance dashboards may provide individuals with a sense of ownership of their work and motivation for achieving goals. Yet, this ‘metrification’ of frontline workers jobs may also dis-empower frontline workers as they strive to meet specific performance and outcome numbers that may be out of their control. In this view, performance metrics may increase administrative supervision of frontline workers while ignoring the larger socioeconomic or contextual factors that may complicate improvements in malnutrition rates. This was a critique of the national health program that ended in 2014, where individual staff on the health teams were evaluated based on strict performance metrics.

The manuscript could be improved by providing more discussion of how the intervention components changed frontline workers performance and subsequently resulted in improvements of the process and outcome measures.

Author Response

Thanks for this very helpful and pointed critique. We agree with all the points raised. To address them, we have done the following:

- Added more text to the Methods (Description of Improvement Intervention) detailing the overall spirit/philosophy guiding the audit/feedback components: collaborative/supportive/problem solving in contrast to punitive/performance oriented

-Added a new Table 1 (room for this made by deprecating the detailed description of the clinical program components to a Supplementary Table) providing additional rational and "example outcomes" for the key components of the improvement intervention

- eliminated use of the word "empowerment" which was somewhat gratuitous in the manuscript as the reviewer points out. Providing support to frontline workers through supervision and feedback is the main point of our intervention, and this is much better detailed now in the new Table 1

-under Results (Monitoring of Process Improvement Indicators) reworded to note that we the process indicators we chose help us to conduct an overall assessment both of improvements in caregiver-worker contact frequency and delivery of essential nutrition components

- Added an additional paragraph to the discussion, emphasizing the importance of collaborative, nonpunitive supervision for effective quality improvement and contrasting this to recent research on supervisory culture in a major Guatemalan hospital. We also, as pointed out by the reviewer,  reference the misuse of inflexible audit culture to defund Guatemala's primary care infrastructure in 2014 and contrast this with our approach here.

Reviewer 2 Report

The study is well-designed and is an important reference point for other researchers. However, I have a few comments:

  1. Issue of editing - space before quoting and dot after quoting, e.g. line 41, 44, 50, 51, 61 etc.
  2. Lines 167, 168 - please replace the dashes with commas.
  3. Table 3 - in the table title - "Data shown for October 2014."
  4. Figure 3 - in the axis title  - A. "Number of children, n"; B. and C. "Proportion of..., %".
  5. Figure 3 - in the chart title - no information on the shifted, dashed horizontal axis.
  6. Figure 4 - in the axis title "Proportion of..., %".
  7. Figure 4 - please define UCL and LCL in Material and Methods section. If ICL / SCL are synonymous with LCL / UCL, please use unified markings.
  8. Figure 5 - in the axis title A. "Z-score, mean", B. "Z-score, standard deviation".

Author Response

  1. Issue of editing - space before quoting and dot after quoting, e.g. line 41, 44, 50, 51, 61 etc

We have made the suggested changes.

  1. Lines 167, 168 - please replace the dashes with commas.

We have made the suggested change.

  1. Table 3 - in the table title - "Data shown for October 2014."

We have made the suggested change.

  1. Figure 3 - in the axis title  - A. "Number of children, n"; B. and C. "Proportion of..., %".

We corrected 3A, 3B and 3C, with the observation that proportion does not have the % sign because it ranges between 0 and 1.

  1. Figure 3 - in the chart title - no information on the shifted, dashed horizontal axis.

We apologize for the omission. The dashed line is the median of the indicator for each time period, which is commonly given on run charts. We have added this to the Figure legend.

  1. Figure 4 - in the axis title "Proportion of..., %". XX

We changed the axis title to "Proportion of children" but did not add the % sign as a proportion ranges between 0 and 1.

  1. Figure 4 - please define UCL and LCL in Material and Methods section. If ICL / SCL are synonymous with LCL / UCL, please use unified markings.

We have made the suggested changes. Figures 4 and 5 were standardized with the terms UCL and LCL, and these were defined in the Material and methods section.

  1. Figure 5 - in the axis title A. "Z-score, mean", B. "Z-score, standard deviation".

We have made the suggested change.

Reviewer 3 Report

In their manuscript “Community-based interventions to reduce child stunting in rural Guatemala: A quality improvement model” the authors present the results of Maya Health Alliance program which was performed in one rural Maya community to improve the impact of interventions aiming to reduce stunting  and chronic child undernutrition.

The article is very interesting but some improvements would, however, be important.

  1. It takes some time to understand whether the authors present a literature review, the results of their work or a summary of different national political interventions. This should be better addressed in the abstract and in Methodology section.
  2. Sometimes is not clear whether the results the authors want to stress concern the contents of the program or results reached by this specific program. If the second, than figure 2 and all similar information should be placed in the methodology section. This should be better addressed.
  3. Although the authors outline some biases and confounding factors, I think that there are certainly other possible confounders that should be stressed not only in the discussion but also in the methodology section. That would be important in order to better correlate the positive outcomes with this specific program and to put in evidence its relevance.

Author Response

It takes some time to understand whether the authors present a literature review, the results of their work or a summary of different national political interventions. This should be better addressed in the abstract and in Methodology section.

This is a good critique and we think the confusion stemmed from a single line in the abstract "Recent literature suggests that one factor limiting impact is inadequate supervisory support for frontline workers" which made it seemed like we were going to offer a comprehensive literature review. This was not our intent, so we've eliminated this line from the abstract. The paper engages with literature on quality improvement and audit/feedback as they arise in the introduction as the context for the work we propose here and comparatively in the discussion, but we do not provide a systematic literature review. We apologize for the misleading sentence in the abstract.

Sometimes is not clear whether the results the authors want to stress concern the contents of the program or results reached by this specific program. If the second, than figure 2 and all similar information should be placed in the methodology section. This should be better addressed.

Thanks for the opportunity to clarify this point. Quality improvement interventions are sometimes difficult to describe because it is important to provide a lot of context for them to be intelligible. In this paper, as we mention in the methodology section, we have closely followed the  the Revised Standards for Quality Improvement Reporting Excellence, which is a checklist commonly accepted for reporting on improvement interventions. According to this checklist, elements such as program design and implementation (Figure 1 and 2 in our manuscript) are usually included under the methodology section, and the Results section is used for reporting on analysis done. This is the approach we have used here and which is also suggested by the reviewer, so we hope this clarification will be sufficient.

Another point which made the methods section tedious is the inclusion of the detailed description of the underlying clinical nutrition program (Table 1) which we included for context but which as the reviewer points out was not the focus of the manuscript. To address this, we have removed this Table and placed it as supplementary material. We've also modified the relevant sentence to emphasize that these details are already published, which helps to focus the reader's attention on the improvement work which is the manuscript's focus. This change was helpful for other reasons, as Reviewer 1 had asked for additional details on the improvement components so we replaced the previous table with a new Table 1 focused on Reviewer 1's request.

Although the authors outline some biases and confounding factors, I think that there are certainly other possible confounders that should be stressed not only in the discussion but also in the methodology section. That would be important in order to better correlate the positive outcomes with this specific program and to put in evidence its relevance.

In the methods section, we have added additional details, including that the time-series GEE approach allows us to perform a sensitivity analysis for autocorrelation between data points due to the including of individual children at multiple time points but also for other time-correlated factors such as seasonality. We also add that we chose this approach because of our interest primarily in the group-level trend. Other analytical approaches (such as mixed linear models) would be more appropriate if our interest was estimating the response of individual children to the intervention and adjusting those estimates for individual-level covariates. We also added additional emphasis to these points in the limitations paragraph in the discussion.

Round 2

Reviewer 1 Report

The authors provide a thoughtful revision of the manuscript based on the comments in the first review. They addressed all of the concerns that I raised, clarifying the content of the improvement intervention, the concepts of 'empowerment' and potential issues of audit culture, particularly in Guatemala, while also helping to clarify the mechanisms through which these changes can influence the outcomes.